# Radioactive Contaminants in Edible Mushrooms: A Comparative Study of ^137^Cs and Natural Radionuclides in Amasya and Tekirdağ, Türkiye

**DOI:** 10.3390/jof11050351

**Published:** 2025-05-01

**Authors:** Afife Akkaya, Sinan Aktaş

**Affiliations:** Department of Biology, Faculty of Science, Selçuk University, 42250 Konya, Turkey; akkayaaffe@gmail.com

**Keywords:** edible mushrooms, ^137^Cs, ^238^U, ^232^Th, ^40^K, bioindicators, comparative analysis, environmental radioactivity, food safety

## Abstract

Mushrooms are a significant component of human diets but can bioaccumulate hazardous substances, including both anthropogenic (^137^Cs) and naturally occurring (^238^U, ^232^Th, and ^40^K) radionuclides. This study quantified these radionuclides in 24 commonly consumed mushroom species collected in Amasya and Tekirdağ, provinces of Türkiye. Using a high-purity germanium (HPGe) detector, we found ^137^Cs activity in the Tekirdağ samples ranging from 3.9 to 127.8 Bq/kg, while the ^137^Cs activity in the Amasya samples ranged from 3.1 to 63.7 Bq/kg. In particular, *Tricholoma terreum* (Tekirdağ) and *Tricholoma imbricatum* (Amasya) exhibited notably higher ^137^Cs concentrations. The concentration of ^238^U varied between 4.8 and 17.5 Bq/kg in the Tekirdağ samples and 6.5 and 16 Bq/kg in the Amasya samples, whereas the ^232^Th and ^40^K values fluctuated across species and regions, with ^40^K sometimes exceeding 1900 Bq/kg. These results highlight that mushrooms can serve as effective bioindicators for residual radioactive contamination and underline the need for periodic monitoring to assess potential public health risks associated with wild mushroom consumption. These findings also offer a valuable dataset for understanding post-Chernobyl fallout dynamics in the forest ecosystems of Türkiye.

## 1. Introduction

Edible mushrooms, particularly those with epigeous (above-ground) fruiting bodies, have played an important role in human diets throughout history [1]. Approximately 2500 edible mushroom species are known worldwide [2], and mushroom cultivation is widespread in many countries, leading to an increase in both the consumption and study of edible fungi [3,4]. While mushrooms offer nutritional and economic benefits, environmental pollution may cause them to accumulate hazardous substances that pose potential health risks for consumers [5].

In particular, long-lived radionuclides in the environment may contribute to serious problems, including genetic mutations and an increased risk of cancer [6]. The 1986 Chernobyl nuclear power plant accident raised global awareness about radioactive contamination and its ecological impacts [7,8]. Mushrooms represent one of the largest biomass components in forest ecosystems, where they can accumulate not only heavy metals but also naturally occurring and artificial radionuclides [9]. This capacity for accumulation is closely tied to their extensive mycelial networks in soil, making them effective bioindicators [3,10,11].

Among the artificial radionuclides of major concern is cesium-137 (^137^Cs), which can persist for decades due to its physical half-life of about 30 years [12,13]. Nuclear weapon tests conducted in the 1950s–1960s, as well as nuclear accidents such as Chernobyl (1986) and Fukushima (2011), have released ^137^Cs into the environment. Additionally, mushrooms can naturally contain ^40^K (potassium-40), ^232^Th (thorium-232), and ^238^U (uranium-238), which they uptake from soil and further introduce into the food chain [2]. Potassium (K), which is essential for cell-volume regulation and pH maintenance, is actively taken up by mushrooms; hence, ^40^K often remains relatively constant in mushroom tissues [14,15,16]. Because ^137^Cs behaves similarly to K, it can be readily absorbed by mushroom tissue.

Following the Chernobyl accident, numerous studies demonstrated that ^137^Cs and related radionuclides can persist in forest soils for extended periods [17,18]. Mushrooms’ direct interaction with soil and their mycelial architecture allow these isotopes to concentrate in mushroom fruiting bodies, which are frequently consumed by humans and wildlife [19,20]. As a result, the regular ingestion of mushrooms with elevated radionuclide levels could increase the risk of internal exposure [12,21].

Given that radiation can arise from both natural (e.g., ^238^U, ^232^Th, ^40^K) and artificial (e.g., ^137^Cs) sources [22,23,24], understanding their combined presence in commonly consumed mushrooms is critical. Therefore, this study focuses on determining the activity concentrations of ^137^Cs and natural radionuclides (^238^U, ^232^Th, and ^40^K) in edible mushroom samples collected in Amasya and Tekirdağ, provinces of Türkiye. We also explore how different species and localities may exhibit varying bioaccumulation potentials. The findings provide valuable data for evaluating potential radiological health risks associated with mushroom consumption and underscore the importance of mushrooms as bioindicators for ecosystem monitoring. The radionuclides studied in this research emit different types of ionizing radiation. Cs-137 and K-40 emit beta and gamma radiation, while U-238 and Th-232 are alpha emitters. Gamma radiation can penetrate biological tissues and pose both internal and external risks. In contrast, alpha particles are less penetrating but highly damaging when inhaled or ingested. Understanding the radiation types and biological risks associated with each isotope is essential for evaluating their environmental and health implications [16].

## 2. Materials and Methods

### 2.1. Study Areas

Amasya is located in the central Black Sea region of Türkiye, between latitudes 41°04′54″ and 40°16′16″ N and longitudes 34°57′06″ and 36°31′53″ E. The area features a transitional Black Sea climate with moderate precipitation levels.

Tekirdağ is situated in the Thracian part of the Marmara Region (40°36′–41°31′ N, 26°43′–28°08′ E). The climate is largely Mediterranean along the coast, whereas inland areas experience partly continental conditions.

### 2.2. Sampling and Sample Preparation

Field surveys were conducted in 2019 in Taşova District (Amasya) and Saray District (Tekirdağ). A total of 24 commonly consumed macrofungal species were collected from each region (Table 1). During collection, visible soil and plant debris were carefully removed from the mushroom samples, and the specimens were placed into sterile bags, labeled, and transported to the laboratory. Species identification was performed by Dr. Sinan AKTAŞ according to standard taxonomic keys.

In addition to mushroom sampling, soil samples were taken from each region at depths of 20–30 cm to determine background radionuclide levels. In the laboratory, mushroom samples were dried at 40–60 °C for three days in a drying oven (etüv). The dried samples were then homogenized using a grinder to ensure uniform particle size.

### 2.3. Preparation of Mushroom Samples and Determination of Radioactivity Content

After drying, the mushroom samples were ground, and sieving was conducted to ensure a uniform particle size. Next, the processed samples were placed into transparent polystyrene containers (6 cm in diameter; 5 cm in height) fitted with white screw caps. The containers were tightly sealed and secured with Parafilm^®^ to prevent any exchange of moisture. Subsequently, they were stored for one month to allow for radioactive equilibrium between the decay products of ^238^U and ^232^Th. Once the storage period was completed, each sample was measured for 50,000 s using a high-purity germanium (HPGe) detector. The acquired spectra were analyzed, and the activity concentrations of the radionuclides were calculated.

The specific activity (A) for each radionuclide was calculated using the following equation:(1)$$A=\frac{Net\;Peak\;Area}{Counting\;Time\times Sample\;Mass\times Abundance\times Detector\;Efficiency}$$

Radioactivity analyses were performed using a multi-channel gamma spectrometer at the Central Research Laboratory of Kastamonu University. Gamma spectroscopic measurements were carried out with an ORTEC GEM50P4-83 high-purity coaxial germanium detector featuring an energy resolution of 1.9 keV at 1332.5 keV and 50% relative efficiency. The detector setup comprised a preamplifier, a spectroscopy amplifier, an analog-to-digital converter (ADC) system, and a multi-channel analyzer (MCA) [25].

Since accurate analysis of the collected spectra requires knowledge of which channels correspond to specific energies, energy calibration was performed using the spectrum of a standard source placed at a fixed distance from the detector. A multi-nuclide standard reference source (or sources) with known emission lines was utilized. For energy and efficiency calibration, point sources containing ^109^Cd, ^57^Co, ^133^Ba, ^22^Na, ^137^Cs, ^54^Mn, and ^60^Co, covering energies from 80 keV to 1400 keV, were used. Table 2 summarizes the primary characteristics of the calibration source [26].

## 3. Results

### 3.1. Radionuclide Activity Concentrations in Tekirdağ Samples

Table 3 presents the activity concentrations of ^238^U, ^232^Th, ^40^K, and ^137^Cs measured in the edible mushroom samples collected in the Tekirdağ region. According to these data, the ^40^K activity was notably higher than that of the other radionuclides in most mushroom species. Moreover, the presence of the artificial isotope ^137^Cs was observed in all samples. Given that ^137^Cs has a physical half-life of 30.17 years, its continued detection suggests residual contamination likely stemming from nuclear fallout, including the Chernobyl accident [1].

In the Tekirdağ samples, ^238^U ranged between 4.8 and 17.5 Bq/kg, ^232^Th ranged between 1.3 and 11.9 Bq/kg, ^40^K ranged between 287.4 and 1940.9 Bq/kg, and ^137^Cs ranged between 3.9 and 127.8 Bq/kg. Notably, the highest ^137^Cs level (127.8 Bq/kg) was recorded in *Tricholoma terreum*, while *Amanita caesarea* exhibited the lowest ^137^Cs concentration (3.9 Bq/kg). These findings highlight significant variability among mushroom species, reflecting differences in both their physiological accumulation capacity and local environmental conditions.

The soil sample from the Tekirdağ site showed activity concentrations of 19.2 Bq/kg for ^238^U, 7.2 Bq/kg for ^232^Th, 362.6 Bq/kg for ^40^K, and 2.3 Bq/kg for ^137^Cs, which are generally lower or comparable to previously reported values in similar regions [2].

Figure 1 illustrates the ^137^Cs activity concentrations for the Tekirdağ samples by species. *Tricholoma terreum* and *Hydnum repandum* stand out with relatively higher ^137^Cs levels, indicating a pronounced capacity to accumulate cesium. Such inter-species differences can be attributed to distinct ecological strategies, mycelial depth, and potassium/cesium ion exchange mechanisms [3].

### 3.2. Radionuclide Activity Concentrations in Amasya Samples

Table 4 summarizes the activities of ^238^U, ^232^Th, ^40^K, and ^137^Cs for mushroom samples from the Amasya region. The measured ^137^Cs activity ranged between 3.1 and 63.7 Bq/kg, with the highest level (63.7 Bq/kg) recorded in *Tricholoma imbricatum*. Meanwhile, the soil sample averaged about 23.1 Bq/kg for ^137^Cs, indicating that local contamination from artificial sources persists in this area. In comparison, ^238^U activity ranged from 6.5 to 16 Bq/kg, ^232^Th activity ranged from 1.3 to 5.7 Bq/kg, and ^40^K activity ranged from 224.7 to 2048.5 Bq/kg.

Figure 2 shows the distribution of ^137^Cs concentrations in the Amasya samples. Similar to the Tekirdağ samples, there is substantial variation among species. For instance, *Hydnum repandum* and *Ganoderma applanatum* typically exhibited moderate ^137^Cs levels, whereas *Tricholoma imbricatum* showed a notably higher uptake. Such differences may arise from varying ecological niches, local soil composition, and the mycelial network’s penetration depth [3].

### 3.3. Comparison of Common Species

Table 5 compares the ^238^U, ^232^Th, ^40^K, and ^137^Cs activity concentrations (Bq/kg) found in common mushroom species collected in the Amasya and Tekirdağ Provinces. Notably, *Macrolepiota procera* from Tekirdağ displayed higher ^238^U (11.9 ± 0.4 Bq/kg) and ^137^Cs (18.2 ± 0.4 Bq/kg) activity concentrations compared to the same species in Amasya (1.3 ± 0.1 Bq/kg for ^238^U and 3.1 ± 0.1 Bq/kg for ^137^Cs). This discrepancy may reflect differences in the local soil uranium content and residual fallout levels. Similarly, *Lactarius deliciosus* and *Hydnum repandum* also presented divergent radionuclide profiles when sampled from each region.

To assess the statistical significance of these discrepancies, Student’s *t*-test (for two groups) or one-way ANOVA (for more than two groups) was applied to the mean activity concentrations, considering *p* < 0.05 an indicator of significance. Marked differences emerged, particularly for ^137^Cs and ^40^K, between the two localities in some species, suggesting that both environmental factors (soil chemistry, altitude, and climate) and species-specific uptake mechanisms contribute to variations [4,6].

### 3.4. Additional Radionuclides and Visual Representations

Figure 3 and Figure 4 show the concentrations of ^232^Th in Tekirdağ and Amasya, respectively, while Figure 5 and Figure 6 present the ^238^U distributions. As both ^232^Th and ^238^U largely derive from soil origins, differences among mushroom species mainly arise from their varying accumulation capabilities and local soil conditions. Figure 7 and Figure 8 illustrate the ^40^K data, indicating that potassium, an essential element for fungal physiology, typically shows higher activity levels. Notably, the ^40^K content often surpasses that of other radionuclides in many mushroom samples, aligning with the literature suggesting that mushrooms tightly regulate potassium [7,8,9].

### 3.5. Comparison with International Studies on ^137^Cs in Mushrooms

A broader comparison with international studies is given in Table 6, demonstrating that ^137^Cs activity in mushrooms can vary substantially across different countries, including Italy, Slovakia, Poland, the Czech Republic, Brazil, and Bulgaria. Seemingly, post-Chernobyl radioactive fallout is still detectable in numerous forest ecosystems worldwide [10,11,27]. Our findings of ^137^Cs levels in Tekirdağ and Amasya (3.1–127.8 Bq/kg) lie within or slightly below the ranges reported in other countries with historical nuclear deposition [12,28]. Notably, Southeastern Turkey appears to be less impacted by ^137^Cs, aligning with historical transport patterns of radioactive clouds after the Chernobyl accident [14,15].

## 4. Discussion

Mushrooms are among the most significant components of forest ecosystems and thus play a key role in studies on the distribution of radioactive elements [3]. Similar to minerals, mushrooms can bind and redistribute radionuclides, which may lead to radiation exposure in humans and animals that consume them. While the radiation to which humans are exposed can derive from both natural (e.g., ^40^K, ^238^U, ^232^Th) and artificial (e.g., ^137^Cs) sources, the present study focused on naturally abundant radionuclides (^40^K, ^238^U, ^232^Th) and the artificial radionuclide ^137^Cs in edible mushroom species. We further compared species- and region-specific differences between the Tekirdağ and Amasya Provinces.

### 4.1. ^40^K and ^137^Cs in Tekirdağ Versus Amasya

As shown in Table 3, the mushroom samples from Tekirdağ generally exhibited higher ^40^K activity than the other radionuclides measured. At the same time, the presence of artificial ^137^Cs was observed in all samples, suggesting that residual contamination from the 1986 Chernobyl accident remains relevant even decades later [8]. Despite ^137^Cs having a physical half-life of about 30 years, its detectability in mushrooms indicates that environmental factors—such as altitude and precipitation—continue to redistribute this radionuclide in forested areas [18]. High-elevation sites often receive more rainfall, which can deposit greater amounts of fallout, thereby elevating ^137^Cs concentrations in the mushrooms growing there.

The Amasya data (Table 4) indicated that the ^137^Cs activity in soil was approximately an order-of-magnitude higher than in Tekirdağ. However, no significant differences were noted between the two regions’ soils with respect to ^232^Th, ^238^U, and ^40^K levels. The mean ^40^K activity in Amasya samples (1158.14 Bq/kg) was slightly higher than in Tekirdağ (1083.66 Bq/kg). Overall, the Tekirdağ mushrooms had higher average concentrations of ^238^U, ^232^Th, and ^137^Cs, whereas the Amasya mushrooms displayed somewhat higher ^40^K concentrations, potentially reflecting differences in local geology, past fallout deposition patterns, and fungal bioaccumulation capacities [9].

### 4.2. Interregional Comparisons of Common Species

Table 5 compares the radionuclide activities of common species from both regions. For instance, *Macrolepiota procera* (Scop.) Singer in Amasya displayed an activity of 6.5 ± 0.2 Bq/kg for ^238^U, whereas the same species in Tekirdağ reached 11.9 ± 0.4 Bq/kg. Such discrepancies could be attributed to a higher ^238^U content in Tekirdağ’s soil or to a species-specific accumulation mechanism. Similarly, the ^232^Th activity in the Amasya samples measured 1.3 ± 0.1 Bq/kg, whereas it reached 7.7 ± 0.2 Bq/kg in the Tekirdağ samples. Meanwhile, the ^137^Cs concentration in *M. procera* ranged from 3.1 ± 0.1 Bq/kg in the Amasya samples to 18.2 ± 0.4 Bq/kg in the Tekirdağ samples, which is consistent with the idea that Tekirdağ may have experienced greater residual fallout [6]. Other common species, such as *Lactarius deliciosus* (L.) Gray and *Hydnum repandum* (L.), also exhibited different ^238^U, ^232^Th, and ^137^Cs levels across regions. These differences in radionuclide accumulation may result from a combination of soil composition and fungal physiology. For example, variations in cation exchange capacity, organic matter, and fungal ion transport specificity can all influence the observed species- and region-dependent uptake [9,16].

An alternative hypothesis could be that in Amasya, radionuclides remain concentrated in the topsoil due to lower leaching, whereas in Tekirdağ, higher rainfall or different soil texture may allow these isotopes to migrate deeper. Consequently, fungi with deeper mycelial systems (e.g., *Macrolepiota procera*) may access and accumulate more radionuclides in Tekirdağ, while surface-colonizing fungi accumulate more in Amasya. Future studies including vertical soil profiling would help validate this hypothesis.

### 4.3. Detailed Observations on ^137^Cs, ^232^Th, and ^238^U

Figure 1 illustrates the distribution of ^137^Cs activity in the Tekirdağ samples, ranging from 3.9 to 127.8 Bq/kg. *Tricholoma terreum* (Schaeff.) P. Kumm had the highest concentration (127.8 Bq/kg), while *Amanita caesarea* (Scop.) Pers recorded the lowest (3.9 Bq/kg). This wide variation underscores the distinct bioaccumulation capacities among fungal taxa [14,15]. In Amasya (Figure 2), the ^137^Cs concentration ranged from 3.1 to 63.7 Bq/kg, with *Tricholoma imbricatum* (Fr.) P. Kumm reaching about 63.7 Bq/kg. Similar findings of high cesium uptake in *Tricholoma* species have been reported [33]. Multiple factors—such as mycelial depth, habitat, forest type, clay content, pH, and microclimate—can influence ^137^Cs accumulation [8,30]. Atmospheric events like rainfall or snowfall may further increase local radioactivity levels [2].

As for ^232^Th (Figure 3 and Figure 4) in Tekirdağ, values ranged between 1.3 and 11.9 Bq/kg, with *Lycoperdon perlatum* Pers. achieving the highest value at 11.9 Bq/kg and *Lactarius deliciosus* (L.) Gray the lowest at 1.3 Bq/kg. The ^238^U concentrations in Tekirdağ (4.8–17.5 Bq/kg) also varied significantly among species, whereas the soil measured 19.2 Bq/kg (Figure 5). Similarly, in Amasya (Figure 6), ^238^U peaked at 16 Bq/kg in *T. imbricatum*, further confirming that both the local soil composition and specific fungus physiology govern accumulation patterns [2].

In this study, ^137^Cs was treated as a purely anthropogenic radionuclide resulting from nuclear accidents, while ^238^U, ^232^Th, and ^40^K were considered part of the natural geogenic background. This classification enables an indirect but effective source attribution framework [2].

### 4.4. ^40^K Accumulation and Homeostasis

Figure 7 and Figure 8 reveal that ^40^K dominates the total radionuclide profile in many samples, with measured values frequently falling in the 1000–2000 Bq/kg range. Because potassium is vital for cellular functions (e.g., osmotic balance, pH regulation), mushrooms have well-developed homeostatic systems for potassium uptake [8,16,34]. Hence, the ^40^K content in mushroom tissues can exceed that in the corresponding soil, as also noted in prior investigations [35].

Mushrooms possess specialized potassium transporters and Na^+^/H^+^ antiporters that enable them to maintain high intracellular K^+^ concentrations even in fluctuating environmental conditions. This homeostasis explains the consistently high levels of K-40 in mushroom tissues and reflects selective ion uptake mechanisms [16,34].

### 4.5. Comparisons with Other Regions and the “Chernobyl Fallout” Map

Correlating our findings with data from other countries (Figure 9, Table 6) suggests that ^137^Cs levels in mushrooms from Italy, Slovakia, Poland, the Czech Republic, Brazil, Bulgaria, and certain regions of Türkiye remain relatively high due to historical nuclear releases [2,17,18,19,20,27,29,30,31,32]. In Figure 9, we present a map showing the “Chernobyl fallout” zone, illustrating how the accident’s radioactive plume affected several parts of Europe, including areas of Türkiye. According to studies by Türkekul et al. [30], ^137^Cs activity tends to be higher in the Black Sea coastal areas than in inland regions of Türkiye, following weather and transport patterns at the time of the accident [6,7]. Even though Türkiye was less impacted by the Chernobyl plume than some Eastern European countries, localized elevated ^137^Cs remains evident in mountainous or high-precipitation environments.

### 4.6. Concluding Remarks

Overall, our data suggest that ^238^U and ^232^Th levels are primarily driven by geological factors, while mushrooms tightly regulate ^40^K. Meanwhile, ^137^Cs levels can vary widely depending on species biology and local contamination histories, illustrating the legacy of major nuclear events such as Chernobyl. Although the levels reported here are generally not alarmingly high, certain species—particularly within the *Tricholoma* genus—could pose an increased internal exposure risk when consumed frequently in large quantities.

Therefore, periodic monitoring of radionuclide content in frequently consumed mushrooms is recommended, particularly in regions known to have experienced fallout or with distinctive geological features. Future studies might incorporate seasonal sampling, dose assessments for local communities, and broader geospatial analyses to clarify radionuclide distribution in Turkish forest ecosystems. These efforts will enhance public health protection while underscoring the role of mushrooms as valuable bioindicators of environmental radioactivity.

Although dose estimation was not within the scope of the current study, future investigations should incorporate internal dose assessments and cancer risk evaluations based on species-specific consumption rates and biokinetic models to better inform public health strategies.

## 5. Conclusions

These findings confirm that, after nuclear accidents such as Chernobyl, certain long-lived radionuclides (particularly ^137^Cs) can persist in forest ecosystems for many years. The main conclusions and recommendations of this study are summarized below.

**Species-Specific Variations**: Some mushroom species show a pronounced capacity for ^137^Cs accumulation, likely due to distinct physiological and chemical properties [14].

**Regional Influences**: In Tekirdağ, several species exhibited more elevated ^137^Cs levels than their counterparts in Amasya, possibly reflecting differences in fallout history and soil composition [6,8].

**Public Health Implications**: Since the routine consumption of mushrooms may lead to increased internal exposure, periodic monitoring of commonly consumed species is advisable [21].

**Role of Mushrooms as Bioindicators**: Mushrooms can serve effectively as bioindicators for identifying nuclear accidents or fallout-induced contamination [7].

In conclusion, our results underscore the importance of regularly monitoring radioactivity levels in frequently consumed mushroom species. The persistence of long half-life artificial isotopes, such as ^137^Cs, raises concerns about internal exposure and necessitates ongoing attention. The data and methodological framework provided by this study can inform future radioecological research. Broader geographic sampling, the inclusion of additional mushroom species, seasonal analyses, and more detailed dose assessments will further clarify how mushrooms can be utilized as comprehensive bioindicators and how to best protect public health.

## Figures and Tables

**Figure 1 jof-11-00351-f001:**
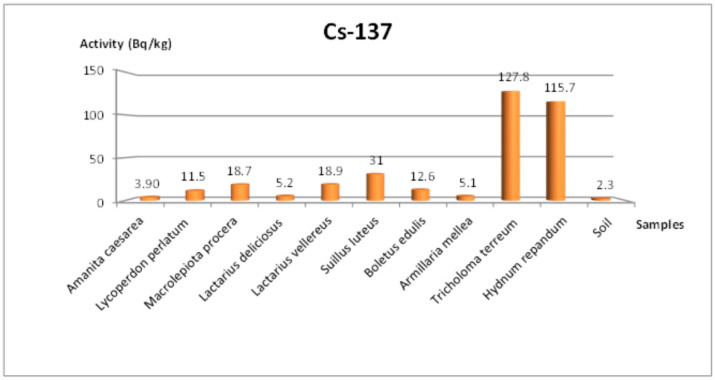
Change in ^137^Cs specific activity concentration according to mushroom species collected from the Tekirdağ region.

**Figure 2 jof-11-00351-f002:**
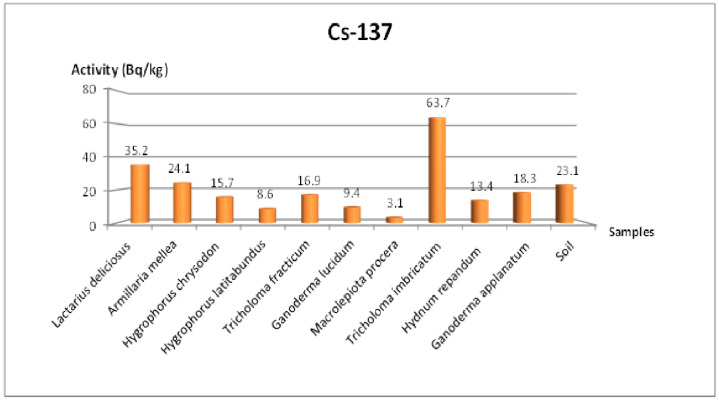
Change in ^137^Cs specific activity concentration according to mushroom species collected in the Amasya region.

**Figure 3 jof-11-00351-f003:**
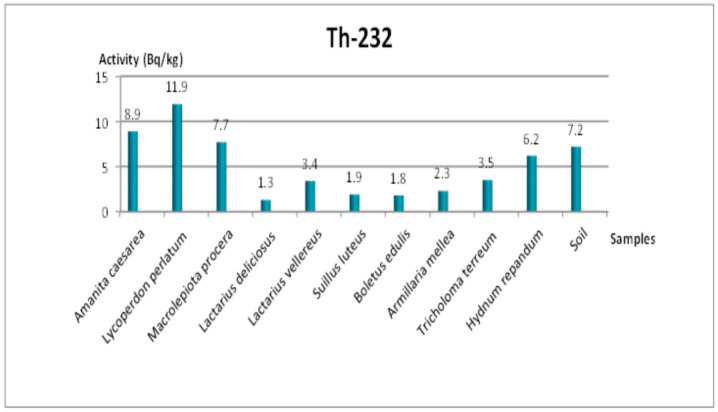
Change in ^232^Th specific activity concentration according to mushroom samples from Tekirdağ.

**Figure 4 jof-11-00351-f004:**
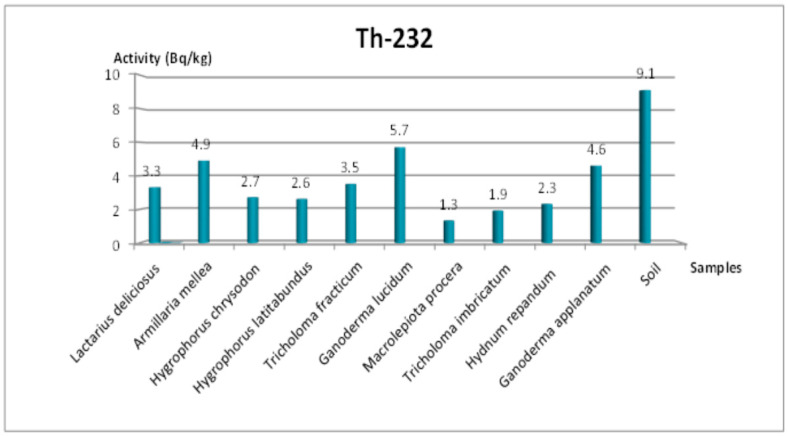
Change in ^232^Th specific activity concentration according to mushroom samples from Amasya.

**Figure 5 jof-11-00351-f005:**
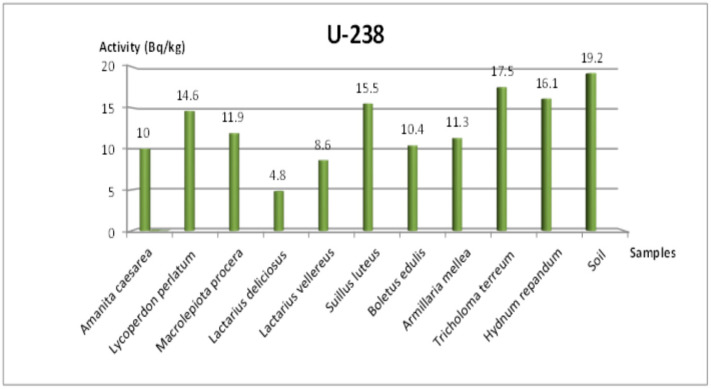
Change in ^238^U specific activity concentration according to mushroom samples from Tekirdağ.

**Figure 6 jof-11-00351-f006:**
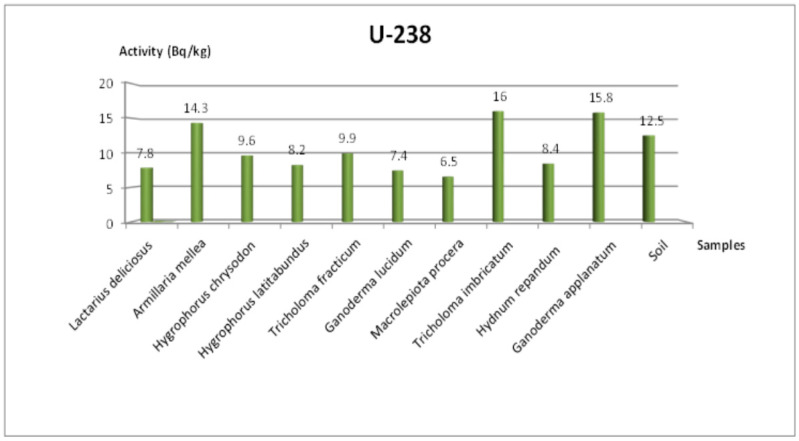
Change in ^238^U specific activity concentration according to mushroom samples from Amasya.

**Figure 7 jof-11-00351-f007:**
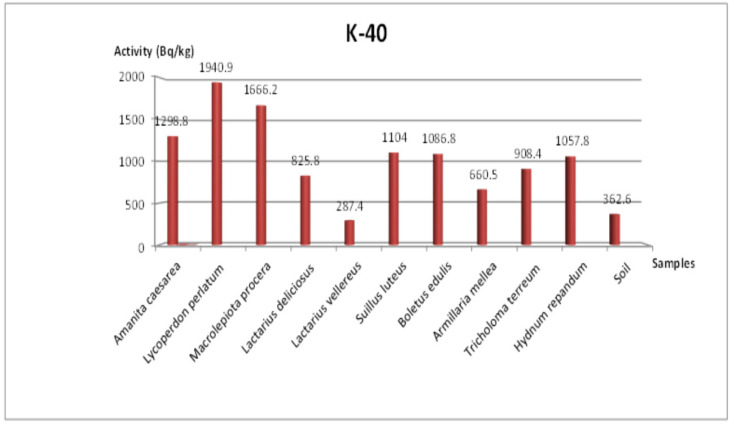
Change in ^40^K specific activity concentration according to mushroom samples from Tekirdağ.

**Figure 8 jof-11-00351-f008:**
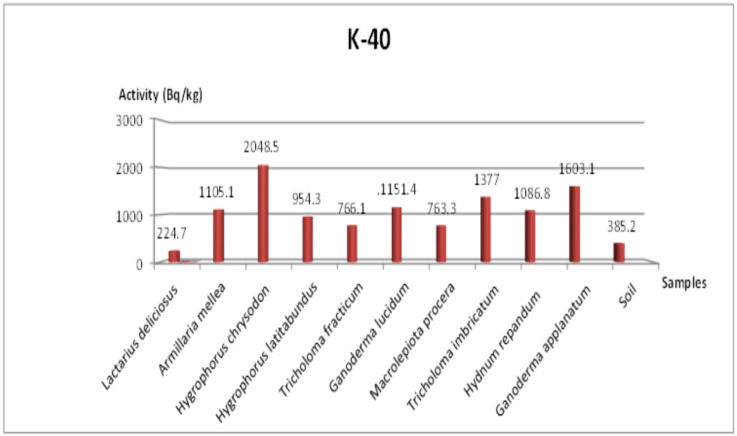
Change in ^40^K specific activity concentration according to mushroom samples from Amasya.

**Figure 9 jof-11-00351-f009:**
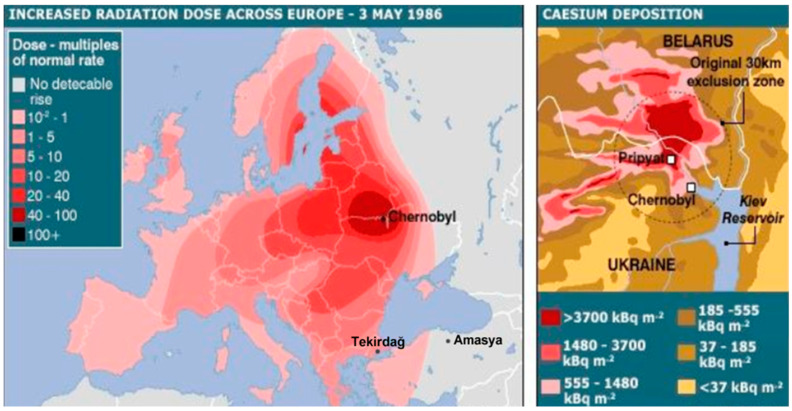
Chernobyl fallout area [36] and study area (Tekirdağ and Amasya).

**Table 1 jof-11-00351-t001:** Edible mushroom species collected in Tekirdağ and Amasya Provinces.

Tekirdağ Province	Amasya Province
*Amanita caesarea* (Scop.) Pers	*Lactarius deliciosus* (L.) Grey
*Lycoperdon perlatum* Pers.	*Armillaria mellea* (Vahl) P. Kumm.
*Macrolepiota procera* (Scop.) Singer	*Hygrophorus chrysodon* (Batsch) Fr.
*Lactarius deliciosus* (L.) Grey	*Hygrophorus latitabundus* Britzelm.
*Lactarius vellereus* (Fr.) Fr.	*Tricholoma fracticum* (Britzelm.) Kreisel
*Suillus luteus* (L.) Roussel	*Ganoderma lucidum* (Curtis) P. Karst
*Boletus edulis* Bull.	*Macrolepiota procera* (Scop.) Singer
*Armillaria mellea* (Vahl) P. Kumm.	*Tricholoma imbricatum* (Fr.) P. Kumm
*Tricholoma terreum* (Schaeff.) P. Kumm	*Hydnum repandum* (L.)
*Hydnum repandum* L.	*Ganoderma applanatum* Pers.

**Table 2 jof-11-00351-t002:** Specifications of standard calibration source.

Isotope	Energy (keV)	Half-Life (Days)	Abundance (%)
^133^Ba	81	3830	33
^109^Cd	88	464	3.72
^57^Co	122.1	271	86
^57^Co	136.5	271	11
^133^Ba	276.4	3830	6.9
^133^Ba	302.8	3830	19
^133^Ba	356	3830	62
^133^Ba	383.8	3830	8.7
^22^Na	511	946	180
^137^Cs	661.6	11,022	85
^54^Mn	834.8	313	100
^60^Co	1173.2	1922	100
^22^Na	1274.5	946	100
^60^Co	1332.5	1922	100

**Table 3 jof-11-00351-t003:** Radionuclide activities of mushroom species in Tekirdağ region (Bq/kg).

Species	^238^U (Bq/kg)	^232^Th (Bq/kg)	^40^K (Bq/kg)	^137^Cs (Bq/kg)
*Amanita caesarea*	10.0 ± 0.3	8.9 ± 0.3	1298.8 ± 51.8	3.9 ± 0.1
*Lycoperdon perlatum*	14.6 ± 0.4	11.9 ± 0.3	1940.9 ± 77.4	11.5 ± 0.3
*Macrolepiota procera*	11.9 ± 0.4	7.7 ± 0.2	1666.2 ± 66.2	18.2 ± 0.4
*Lactarius deliciosus*	4.8 ± 0.1	1.3 ± 0.1	825.8 ± 32.9	5.2 ± 0.2
*Lactarius vellereus*	8.6 ± 0.3	3.4 ± 0.1	287.4 ±11.1	18.9 ± 0.4
*Suillus luteus*	15.5 ± 0.4	1.9 ± 0.1	1104.0 ± 42.5	31.0 ± 0.8
*Boletus edulis*	10.4 ± 0.3	1.8 ± 0.1	1086.8 ± 41.6	12.6 ± 0.4
*Armillaria mellea*	11.3 ± 0.3	2.3 ± 0.1	660.5 ± 26.1	5.1 ± 0.1
*Tricholoma terreum*	17.5 ± 0.4	3.5 ± 0.1	908.4 ± 35.8	127.8 ± 3.3
*Hydnum repandum*	16.1 ± 0.4	6.2 ± 0.2	1057.8 ± 42.1	115.7 ± 2.8

**Table 4 jof-11-00351-t004:** Radionuclide activities (Bq/kg) in mushroom species collected from the Amasya region.

Species	^238^U (Bq/kg)	^232^Th (Bq/kg)	^40^K (Bq/kg)	^137^Cs (Bq/kg)
*Lactarius deliciosus*	3.3 ± 0.1	7.8 ± 0.2	224.7 ± 8.9	35.2 ± 0.9
*Armillaria mellea*	4.9 ± 0.1	14.3 ± 0.4	1105.1 ± 44.1	24.1 ± 0.3
*Hygrophorus chrysodon*	2.7 ± 0.1	9.6 ± 0.3	2048.5 ± 81.5	12.7 ± 0.4
*Hygrophorus latitabundus*	2.6 ± 0.1	8.2 ± 0.3	954.3 ± 38.4	8.6 ± 0.2
*Tricholoma fracticum*	3.5 ± 0.1	9.9 ± 0.3	766.1 ± 30.2	16.9 ± 0.4
*Ganoderma lucidum*	5.7 ± 0.2	7.4 ± 0.2	1151.4 ± 46.1	9.4 ± 0.2
*Macrolepiota procera*	1.3 ± 0.1	6.5 ± 0.2	763.3 ± 30.2	3.1 ± 0.1
*Tricholoma imbricatum*	1.9 ± 0.1	16.0 ± 0.5	1377.0 ± 54.7	63.7 ± 1.6
*Hydnum repandum*	2.3 ± 0.1	8.4 ± 0.3	1587.9 ± 63.3	13.4 ± 0.4
*Ganoderma applanatum*	4.6 ± 0.1	15.8 ± 0.4	1603.1 ± 63.8	18.3 ± 0.4
Average	3.28	10.39	1158.14	20.54
Soil	9.1 ± 0.3	12.5 ± 0.3	385.2 ± 15.2	23.1 ± 0.5

**Table 5 jof-11-00351-t005:** ^238^U, ^232^Th, ^40^K, and ^137^Cs activity concentrations (Bq/kg) in common mushroom species collected in the Amasya and Tekirdağ Provinces.

Species	^238^U (Bq/kg)	^232^Th (Bq/kg)	^40^K (Bq/kg)	^137^Cs (Bq/kg)	Locality
*Macrolepiota procera*	6.5 ± 0.2	1.3 ± 0.1	763.3 ± 30.2	3.1 ± 0.1	Amasya
*Macrolepiota procera*	11.9 ± 0.4	7.7 ± 0.2	1666.2 ± 66.2	18.2 ± 0.4	Tekirdağ
*Lactarius deliciosus*	7.8 ± 0.2	3.3 ± 0.1	224.7 ± 8.9	35.2 ± 0.9	Amasya
*Lactarius deliciosus*	4.8 ± 0.1	1.3 ± 0.1	825.8 ± 32.9	5.2 ± 0.2	Tekirdağ
*Hydnum repandum*	8.4 ± 0.3	2.3 ± 0.1	1587.9 ± 63.3	13.4 ± 0.4	Amasya
*Hydnum repandum*	16.1 ± 0.4	6.2 ± 0.2	1057.8 ± 42.1	115.7 ± 2.8	Tekirdağ
Soil	12.5 ± 0.3	9.1 ± 0.3	385.2 ± 15.2	23.1 ± 0.5	Amasya
Soil	19.2 ± 0.5	7.2 ± 0.2	362.6 ± 14.3	2.3 ± 0.1	Tekirdağ

**Table 6 jof-11-00351-t006:** ^137^Cs activity concentration studies conducted in other countries.

Countries	137Cs (Bq/kg)	Year	Reference
Italy	10.33–732.29	2001	[27]
Slovakia	322.9–869.6	2005	[17]
Poland	330–16,670	2006	[29]
Slovakia	2.4–720	2006	[18]
Czech Rep.	0.4–708	2006	[18]
Türkiye	27.2–28.4	2007	[20]
Brazil	1.45–10.6	2012	[2]
Türkiye (Ordu-Kastamonu)	-	2014	[19]
Türkiye (Orta Karadeniz)	<0.01–697	2018	[30]
Türkiye (Trabzon)	-	2019	[31]
Bulgaria	98–124	2020	[32]
Türkiye (Amasya-Tekirdağ)	3.1–127.8	2021	This study

## Data Availability

The data presented in this study are not publicly available due to privacy and confidentiality restrictions. However, specific details regarding the dataset and experimental methodology have been provided in the manuscript to ensure transparency and reproducibility. Researchers who require further information may contact the corresponding author, subject to compliance with applicable ethical, institutional, and legal constraints.

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
