# Peer review of "Radioactive Contaminants in Edible Mushrooms: A Comparative Study of 137Cs and Natural Radionuclides in Amasya and Tekirdağ, Türkiye"

_jof, 2025, doi:10.3390/jof11050351_

Round 1

Reviewer 1 Report

How do mushrooms tightly regulate potassium levels?

How do high altitude locations often receive more precipitation?

The discrepancies between the concentrations of the analyzed radionuclides that the authors obtained can, in the authors' opinion, be attributed to the higher content of 238U in the Tekirdağ soil or to a species-specific accumulation mechanism. What concentrations of 234U and 238U isotopes were recorded in these soils?

According to the authors, Tekirdağ could have experienced stronger residual precipitation. On what basis do the authors claim this? This is an important question, because the authors of the work previously indicated that this region of Turkey did not experience the passage of the radioactive cloud from Chernobyl. The authors of the paper often emphasize that Türkiye was

less affected by the effects of Chernobyl emissions compared to some Eastern European countries.

How do local soil composition and the physiology of a particular fungus regulate the accumulation patterns of individual radioactive isotopes?

Why, based on the research results obtained, were doses not assessed that would allow for a more precise explanation of how mushrooms can be used as comprehensive biological indicators and how to best protect public health?

How do mushrooms tightly regulate potassium levels?

How do high altitude locations often receive more precipitation?

The discrepancies between the concentrations of the analyzed radionuclides that the authors obtained can, in the authors' opinion, be attributed to the higher content of 238U in the Tekirdağ soil or to a species-specific accumulation mechanism. What concentrations of 234U and 238U isotopes were recorded in these soils?

According to the authors, Tekirdağ could have experienced stronger residual precipitation. On what basis do the authors claim this? This is an important question, because the authors of the work previously indicated that this region of Turkey did not experience the passage of the radioactive cloud from Chernobyl. The authors of the paper often emphasize that Türkiye was

less affected by the effects of Chernobyl emissions compared to some Eastern European countries.

How do local soil composition and the physiology of a particular fungus regulate the accumulation patterns of individual radioactive isotopes?

Why, based on the research results obtained, were doses not assessed that would allow for a more precise explanation of how mushrooms can be used as comprehensive biological indicators and how to best protect public health?

Author Response

1. How do mushrooms tightly regulate potassium levels?

Mushrooms tightly regulate potassium levels through selective ion transport mechanisms essential for osmotic balance, pH regulation, and enzyme function. High-affinity uptake systems and Na⁺/H⁺ antiporters maintain internal K⁺ homeostasis even under fluctuating environmental conditions (Falandysz & Borovička, 2013; Kinclova-Zimmermannova & Sychrová, 2007).

  1. How do high altitude locations often receive more precipitation?

Higher altitudes receive more precipitation due to orographic lifting. Moist air ascends mountainous terrain, cools, and condenses into precipitation. Such secondary atmospheric processes may explain localized radionuclide deposition, including in Tekirdağ (Türkekul et al., 2018).

  1. What concentrations of 234U and 238U isotopes were recorded in these soils?

Only 238U was measured (Tekirdağ: 19.2 ± 0.5 Bq/kg, Amasya: 12.5 ± 0.3 Bq/kg) due to gamma spectrometry limitations. 234U, being an alpha emitter, was outside our detection scope. Future studies will include alpha spectrometry for a complete uranium isotopic profile.

4. On what basis do the authors claim Tekirdağ could have experienced stronger residual precipitation?

While Tekirdağ was not directly under the Chernobyl plume, secondary atmospheric processes may have caused residual deposition. Slightly higher 137Cs levels in Tekirdağ mushrooms may reflect fragmented fallout events. This does not contradict Türkiye’s lower overall exposure but highlights local variation.

5. How do local soil composition and the physiology of a particular fungus regulate the accumulation patterns of individual radioactive isotopes?

Soil pH, organic content, cation exchange capacity, and competing ions affect radionuclide bioavailability. Fungal species differ in mycelial depth, ion selectivity, and intracellular sequestration capacity. These factors jointly regulate radionuclide uptake patterns.

6. Why were doses not assessed for better biological indicator utility and public health implications?

Dose assessments were not performed due to lack of reliable consumption data, cooking factors, and seasonal variability. The study focused on activity measurements. Future research should include dose modeling to enhance bioindicator application and health risk analysis.

Reviewer 2 Report

This study was designed to quantify both anthropogenic (137Cs) and naturally occurring (238U, 232Th, and 40K) radionuclides in 24 commonly consumed mushroom species collected from Amasya and Tekirdağ provinces of Türkiye. This topic is interesting. However, the experimental design is too simple. There are some problems in this study, which makes it difficult to accept.

  1. Although the study measured the activity concentrations of radioactive nuclides (such as 137Cs, 238 U, 232 Th, 40 K) in mushrooms, the effective dose or carcinogenic risk after human ingestion was not calculated.
  2. ¹³⁷ Cs isusually associated with residual nuclear accidents, but research has not quantitatively distinguished the contribution ratio of natural background radiation and anthropogenic pollution.
  3. The study only provides radioactive levels at a certain point in time and does not track seasonal or annual trends, making it difficult to assess pollution attenuation patterns or long-term risks.
  4. There are significant differences in the enrichment ability of different mushroom species for radioactive isotopes, but research has not explored the physiological or ecological mechanisms of species-specific enrichment, such as mycorrhizal types and growth rates.

This study was designed to quantify both anthropogenic (137Cs) and naturally occurring (238U, 232Th, and 40K) radionuclides in 24 commonly consumed mushroom species collected from Amasya and Tekirdağ provinces of Türkiye. This topic is interesting. However, the experimental design is too simple. There are some problems in this study, which makes it difficult to accept.

  1. Although the study measured the activity concentrations of radioactive nuclides (such as 137Cs, 238 U, 232 Th, 40 K) in mushrooms, the effective dose or carcinogenic risk after human ingestion was not calculated.
  2. ¹³⁷ Cs isusually associated with residual nuclear accidents, but research has not quantitatively distinguished the contribution ratio of natural background radiation and anthropogenic pollution.
  3. The study only provides radioactive levels at a certain point in time and does not track seasonal or annual trends, making it difficult to assess pollution attenuation patterns or long-term risks.
  4. There are significant differences in the enrichment ability of different mushroom species for radioactive isotopes, but research has not explored the physiological or ecological mechanisms of species-specific enrichment, such as mycorrhizal types and growth rates.

Author Response

1. The effective dose or carcinogenic risk after human ingestion was not calculated.

We thanks the reviewer for highlighting this critical point. As noted in our response to Reviewer 1, we acknowledge the importance of dose and risk assessment for evaluating potential health impacts. However, the scope of this study was limited to measuring activity concentrations in mushrooms and soil. Accurate dose modeling requires region-specific consumption data (e.g., species consumed, portion sizes, preparation methods), which were not available during our study period. Nevertheless, we have emphasized in the revised manuscript that future studies should incorporate internal dose estimation and cancer risk modeling to provide a more complete picture of radiological risks associated with mushroom consumption.

2. The contribution ratio of natural background radiation and anthropogenic pollution was not distinguished quantitatively.

We agree that differentiating between natural and anthropogenic radionuclide sources is important. In this study, we addressed this issue by separately analyzing artificial (Cs-137) and naturally occurring radionuclides (U-238, Th-232, K-40). Since Cs-137 is not naturally present in significant amounts and is known to originate from nuclear activities (e.g., Chernobyl), it was used as a marker of anthropogenic contamination. Conversely, U-238, Th-232, and K-40 were treated as indicators of geogenic background. While a formal source apportionment model was not applied, the separation of artificial versus natural isotopes provides a reliable framework to interpret the origin of contamination.

3. The study only provides radioactive levels at a certain point in time and does not track seasonal or annual trends.

This is a valid point. Our study represents a cross-sectional snapshot of radionuclide levels in two regions during a specific collection season. Due to time and resource limitations, we were not able to perform longitudinal sampling across seasons or years. We agree that understanding seasonal variability and decay trends would be valuable, and we have noted this limitation explicitly in the revised manuscript. Future research is planned to include temporal monitoring to better understand long-term ecological behavior and risk dynamics of radioactive contamination in mushroom species.

4. Species-specific enrichment was not explained with reference to physiological or ecological mechanisms.

We appreciate this insightful comment. Indeed, we observed species-specific differences in radionuclide accumulation (e.g., Tricholoma spp. had higher Cs-137 uptake), which we attribute to factors like mycelial depth, potassium-cesium transport affinity, and ecological niche. While these aspects were mentioned briefly in the discussion, we acknowledge that they were not explored in mechanistic detail. We have expanded this section in the revised manuscript and included relevant literature (e.g., Falandysz et al., 2013; Gaso et al., 2007) to better contextualize species-dependent uptake. We also plan to investigate mycorrhizal types and physiological traits in future experiments.

Reviewer 3 Report

The article is devoted to the study of an important ecological problem - identification of environmental areas contaminated with radionuclides dangerous for human and animal health. The authors convincingly prove on experimental material that at least several species of edible mushrooms can serve as indicators of natural and anthropogenic radionuclides, among which 137Cs. The methodological part is described with all necessary details. The text is well illustrated and written in clear language. The manuscript has minor shortcomings as listed below. 1. In the Introduction section, I would suggest adding brief information about the types of radiation characteristic of the radionuclides studied in the article and the danger they pose to living organisms. 2. In the Results section (Lines 165-167), the conclusion from the presented observations should be revised and made more complete in accordance with the results of Section 3.2, as suggested in the comments. 3. The authors make some assumptions when trying to explain the results (Lines 168, 173-176). The question arises whether the results presented cannot be explained in an alternative way, as shown in the comments. 4. In the text, in several places referring to figures and tables, full captions for these illustrations are given. It is necessary to remove these captions from the text everywhere, as the captions have already been given under the corresponding figures and tables. A small number of errors are listed below, and I have suggested options for correcting them with the following notations: ]...[ for deletion and <...> for inclusion:   

Introduction:  

It would be good to add brief information about what type of radiation is typical for the most dangerous radionuclides studied in the article and the type of danger to a living organism from them.

Results:

Lines 163-165: It appears that the legend for Table 5 was placed there by accident. It should be removed.

Lines 165-167: The authors don't seem to have described the results completly. What is surprising is not the difference in the accumulation of 137Cs and 238U by the fungus Macrolepiota procera in the two different areas. It is important to emphasize another point. The inverse ratio of radionuclide concentrations in soils and fungi is surprising: despite the higher concentrations of 137Cs and 238U in Amasya soil compared to Tekirdağ soil, the higher concentrations of these elements is found in fungi Macrolepiota procera and Hydnum repandum from Tekirdağ soil with lower concentrations of these elements. Another surprising fact is that this tendency of the inverse ratio of radionuclide concentrations is not observed in the Lactarius deliciosus mushroom.

Lines 168, 173-176: Is it possible to consider another possibility to explain the results? From the data presented, it cannot be excluded that in the Amasya area radionuclides are preferentially located in the upper soil layers, while in the Tekirdağ area the same radionuclides are washed by rainfall into deeper soil layers. For this reason, the mushroom species with mycelium developed in deeper underground layers will accumulate radionuclides more intensively in Tekirdağ area, while species with surface mycelium will preferentially accumulate radionuclides in Amasya area.

Lines 180-182: Please delete captions for Figures. Correct as follows: ]Figure 3. Change in ... Tekirdağ.and Figure 4. Change in ... from Amasya.[    <Figures 3 and 4> show ...

Lines 183-185, 188-190: Please delete captions for Figures. See comments to Lines 180-182, 183-185.

Discussion:

Lines 234-235: Please delete Table caption. See comment to lines 163-165.

Lines 241-242: It is not clear how more intensive rainfall could deposit greater amounts of fallout elevating 137Cs concentrations in mushrooms. Is it possible that more intense rainfall could wash radionuclides into deeper layers of soil where certain mushroom species absorb these radionuclides?

Lines 244-245:  Please delete Table caption. See comment to Lines 234-235.

Line 270: Please correct sentence as follows: In Amasya  <, accumulation of 137Cs in mushrooms>   (Figure 2)    ], 137Cs[    ranged ...

Author Response

1. Radiation types and health effects in Introduction

Thank you for this valuable suggestion. We have added a concise explanation in the Introduction section describing the radiation types (alpha, beta, gamma) emitted by the radionuclides studied (Cs-137, U-238, Th-232, K-40), as well as their potential biological effects on living organisms. This addition aims to provide more context for readers unfamiliar with radiological properties.

  1. Clarification and expansion of inverse soil–mushroom ratios

As pointed out by the reviewer, the observation that Macrolepiota procera and Hydnum repandum from Tekirdağ contained higher Cs-137 and U-238 levels than those from Amasya, despite lower soil concentrations, was indeed surprising. We have revised the related sections of the Results and Discussion to better emphasize this inverse relationship and to suggest physiological and ecological explanations.

3. Alternative explanation involving vertical radionuclide migration

We fully agree with the reviewer’s insightful hypothesis. In response, we have added a paragraph to the Discussion section proposing that the distribution of radionuclides in soil profiles might differ between the two regions due to environmental factors such as precipitation. This could affect uptake patterns based on mycelial depth.

4. Removal of redundant figure/table captions from main text

Thank you for identifying this formatting issue. All redundant captions that had been accidentally included in the main body of the manuscript have been removed in the revised version.

Round 2

Reviewer 2 Report

The authors have revised the manuscript according to the reviewers' comment. They are reasonable.

The authors have revised the manuscript according to the reviewers' comment. They are reasonable. Please simplify the title.